# Socioeconomic and lifestyle factors associated with hearing loss in older adults: a cross-sectional study of the English Longitudinal Study of Ageing (ELSA)

Dialechti Tsimpida,[1] Evangelos Kontopantelis,[2] Darren Ashcroft,[3] Maria Panagioti[4]

For numbered affiliations see end of article.

**Correspondence to**
Ms Dialechti Tsimpida;
dialechti.tsimpida@manchester.ac.uk

## ABSTRACT

**Objectives** Aims were (1) to examine whether socioeconomic position (SEP) is associated with hearing loss (HL) among older adults in England and (2) whether major modifiable lifestyle factors (high body mass index, physical inactivity, tobacco consumption and alcohol intake above the low-risk-level guidelines) are associated with HL after controlling for non-modifiable demographic factors and SEP.

**Setting** We used data from the wave 7 of the English Longitudinal Study of Ageing, which is a longitudinal household survey dataset of a representative sample of people aged 50 and older.

**Participants** The final analytical sample was 8529 participants aged 50–89 that gave consent to have their hearing acuity objectively measured by a screening audiometry device and did not have any ear infection.

**Primary and secondary outcome measures** HL defined as >35 dBHL at 3.0 kHz (better-hearing ear). Those with HL were further subdivided into two categories depending on the number of tones heard at 3.0 kHz.

**Results** HL was identified in 32.1% of men and 22.3% of women aged 50–89. Those in a lower SEP were up to two times more likely to have HL; the adjusted odds of HL were higher for those with no qualifications versus those with a degree/higher education (men: OR 1.87, 95%CI 1.47 to 2.38, women: OR 1.53, 95%CI 1.21 to 1.95), those in routine/manual occupations versus those in managerial/professional occupations (men: OR 1.92, 95%CI 1.43 to 2.63, women: OR 1.25, 95%CI 1.03 to 1.54), and those in the lowest versus the highest income and wealth quintiles (men: OR 1.62, 95%CI 1.08 to 2.44, women: OR 1.36, 95%CI 0.85 to 2.16, and men: OR1.72, 95%CI 1.26 to 2.35, women: OR 1.88, 95%CI 1.37 to 2.58, respectively). All regression models showed that socioeconomic and the modifiable lifestyle factors were strongly associated with HL after controlling for age and gender.

**Conclusions** Socioeconomic and lifestyle factors are associated with HL among older adults as strongly as core demographic risk factors, such as age and gender. Socioeconomic inequalities and modifiable lifestyle behaviours need to be targeted by the health policy strategies, as an important step in designing interventions for individuals that face hearing health inequalities.

### Strengths and limitations of this study

► The first study that focuses on modifiable lifestyle factors (such as high body mass index, physical inactivity, tobacco consumption and alcohol intake above the low-risk-level guidelines) associated with hearing loss (HL) among older adults in England.

► Examines four different socioeconomic position (SEP) indicators to HL (education, occupation, income and wealth), instead of a proxy measure to reflect one's total SEP, capturing, therefore, most of the variation in socioeconomic stratification, to the objectively measured HL in older adults.

► The analyses were based on a representative cohort of 8529 participants contained in English Longitudinal Study of Ageing (ELSA), which is a rich resource of information on the dynamics of health, social, well-being and economic circumstances of the English population aged 50 and over.

► The ELSA dataset did not contain information concerning the occupational and social noise exposure, but we examined the association of manual occupations with HL and its attenuation by modifiable determinants including smoking habit, which is of a higher prevalence among those that work in routine and manual occupations in England.

► All the analysed factors explained less than one-third of the variance for the prevalence of HL suggesting that there are additional major factors associated with HL in older adults which have not been included in our analyses.

## INTRODUCTION

Hearing loss (HL) is a major global health challenge and the most prevalent sensory disorder. Approximately 15% of the global adult population has some degree of HL (of at least ≥25 dB HL in the better-hearing ear)[1] and almost 7% has disabling HL (defined as a hearing threshold ≥40 dB HL in the better ear).[2] HL has negative physical, social,

BMJ

cognitive, economic and emotional consequences and is the fourth leading contributor to years lived with disability worldwide.[2]

Previous studies have reported that HL increases with age,[3] exposure to high occupational and social noise[4] and is more commonly in men.[3] There is growing evidence that there are a number of modifiable risk factors for HL,[56] and, if eliminated, half cases of HL could be prevented.[2] Thus, there is a high potential for reducing the burden of HL, if we understand the modifiable factors and the mechanisms that lead to hearing health inequalities, which—following the glossary for health inequalities[7]—could be defined as the avoidable differences in people's hearing health across different social and/or population groups.

Prior research has established health disparities in a wide range of health conditions according to socioeconomic position (SEP).[8] Furthermore, there is an evidence that several modifiable lifestyle factors, such as smoking,[9] alcohol consumption,[10] high body mass index (BMI) and physical inactivity[11] are associated with hearing health. Of course, causal paths have not been established, and these associations may be confounded by deprivation or aspects of deprivation (eg, type of occupation). Nevertheless, quantifying such associations is the first step in that direction; hearing health inequalities is an emerging research area and the existing evidence on the relationship of HL with SEP and modifiable lifestyle factors is scarce. There is a major public health need to assess whether HL is associated with SEP and lifestyle factors because this understanding could inform recommendations for HL preventative strategies. These could include wider implementation of interventions to promote 'healthier lifestyles', or governmental policies for socioeconomic equity among older people in the community.

The aims of this study were (1) to examine whether SEP is associated with HL among older adults in England and (2) whether major modifiable lifestyle factors are associated with HL after controlling for non-modifiable demographic factors and SEP in the analyses. This study is the first that examines four different SEP indicators (education, occupation, income and wealth) in HL, encompassing thus aspects of the life-course socioeconomic stratification,[12] to the objectively measured HL in older adults. In addition, this is the first study that explores how major lifestyle factors for general health outcomes in the English population aged 50 years old and above (such as smoking, high BMI, insufficient physical activity, tobacco consumption and alcohol intake above the low-risk-level guidelines),[1314] account for the variance in HL.

## METHODS
### Study population
The present study used data from the English Longitudinal Study of Ageing (ELSA). The ELSA is a longitudinal household survey dataset of a representative sample of people aged 50 and older in England. It is designed as a large-scale prospective cohort study, with repeat measures

of core variables over numerous waves, in order to explore trajectories on the health, social, well-being and economic circumstances.[15] The current sample contains data from up to eight waves of data collection covering a period of 15 years, with an ongoing 2-year follow-up longitudinal design.[16]

Objective hearing health data were available only in wave 7, where information was collected from 9666 participants, between June 2014 and May 2015. For the purposes of this study, the final analytical sample was n=8529 participants, aged 50–89, that gave consent to have their hearing acuity measured by a screening audiometry device and did not have any ear infection or a cochlear implant.

### Patient and public involvement
Patients were not involved in the conduct of the study.

### Hearing test
A handheld audiometric screening device (HearCheck)[17] was used for the objective measurement of hearing acuity. This is a portable and easy-to-use hearing screening test by Siemens that tests for audibility of pure tone beeps, according to the number of tones that the respondent can hear for each sequence (at 1.0 kHz and 3.0 kHz), per each ear. The functional test sequence begins with a series of three sounds, which have decreasing volume at 1.0 kHz (55, 34 and 20 dB HL) and afterwards another three sounds with decreasing volume at 3.0 kHz (75, 55 and 35 dB HL). Prerequisites for the test were the device to make proper contact with the ear that is tested, hearing aid(s), glasses, earrings and hair bands to be removed to prevent from getting in the way of the hearing device and the room to be as quiet as possible. Participants indicated when they hear the sound by raising their finger. The total number of tones that the participants indicated they could hear in the sequence of sounds at 1.0 and 3.0. kHz, per each ear, was recorded and the total tones heard in the better-hearing ear used for the categorisation of those with HL.

Previous studies have assessed the accuracy of the Siemens HearCheck in detecting HL and compared it with pure tone air conduction averages designated as gold-standard values. Fellizar-Lopez *et al* found that in cases of moderate or worse HL, the HearCheck test fulfils all criteria of high sensitivity rate, high specificity rate and high positive predictive values to be considered an accurate tool to screen for HL, without the need for sound-proof audiometry booths.[18]

### Outcomes
#### Hearing loss
HL was defined as >35 dB HL at 3.0 kHz, in the better-hearing ear. Those with HL were further subdivided into two categories depending on the number of tones heard at 3.0 kHz. This is the level where intervention for HL has shown to be definitely beneficial.[19] For that reason, this categorisation has previously been used in the literature for the characterisation of those assessed by the same

audiometric screening device (HearCheck).[6] Thus, we further explored potential differences in the association between SEP indicators and HL, according to the severity of HL, as measured by HearCheck. The categorisation of those with HL was as following:

1. 'Moderate HL': tones heard at 75 and 55 dB HL but not at 35 dB HL (the first two of the three tones at 3.0 kHz heard),
2. 'Moderately severe or severe HL': tone heard or not at 75 dB HL and tones not heard at 55 dB HL and 35 dB HL (0 or 1 of the three tones at 3.0 kHz heard).

The ordinal variable 'hearing acuity' (in the better-hearing ear) was consisted of the above two categories of HL and the category of 'normal hearing', which was defined as having heard all the three tones of the hearing screening test at 3.0 kHz.

## Indicators of SEP

Education, occupation, income and wealth were the four selected indicators of SEP and information was collected in the seventh wave of ELSA, between June 2014 and May 2015. We considered five categories of the highest educational attainment: degree/higher education; A level (Level 3 of the National Qualifications Framework); O levels Certificate of Secondary Education; foreign/other; no qualifications. Tertiles of self-reported occupation were based on the National Statistics Socio-economic Classification: managerial and professional, intermediate, routine and manual occupations). The relative financial position of the participants was captured by quintiles of the net household income (first quintile lowest; fifth quintile highest) that is summed across household members. In order to avoid the information bias that is related to the retirement status, we used quintiles of the total non-pension wealth that is reported at the household level (first quintile lowest; fifth quintile highest), which represents the sum of net financial wealth, net physical wealth and net housing wealth.

## Covariates

Age, marital status, retirement status and non-medical determinants of health (BMI, physical activity, tobacco and alcohol consumption) were assessed as covariates in the association between SEP indicators and HL.[5]

Age was categorised into three groups (50–64, 65–74 and 75–89), to allow for a comparison with Benova et al,[20] who examined the association of SEP with self-reported hearing difficulty in ELSA wave 2.

Marital status was dichotomised into those that are currently married or not. Those who are currently married included the categories (1) married, first and only marriage, (2) in a registered civil partnership and (3) remarried, in a second or later marriage. Those that categorised as not currently married included the categories (1) single, that is never married and never registered in a marriage, (2) separated, but still legally married, (3) divorced and (4) widowed.

Retirement status was dichotomised into those who were retired or not, according to the self-reported employment status.

BMI measurements were grouped in four categories, according to WHO definitions[21]: (1) underweight: BMI under 18.5 $kg/m^2$, (2) normal: BMI 18.5 $kg/m^2$ or over but less than 25 $kg/m^2$, (3) overweight: BMI 25 $kg/m^2$ or over but less than 30 and (4) obese: BMI 30 $kg/m^2$ or over.

Tobacco consumption of any type of nicotine products was recoded into three categories: those that were current smokers, those that were former smokers and those that never smoked. Both current and former smokers answered the question of 'number of cigarettes smoked per day', to explore whether they were occasional or regular smokers.

Alcohol consumption was recorded using several continuous variables such as the number of days of alcohol consumption in the last 7 days and the number of (1) measures of spirit, (2) glasses of wine and (3) pints of beer that the respondents had consumed during this period. We constructed a continuous variable to represent the sum of units of alcohol that the participants consumed in the last 7 days, according to the Chief Medical Officer's Drinking Guidelines[22] that counts as 1 unit each measure of spirit and as 2 units each glass of wine of pint of beer. The constructed variable of units of alcohol during the last 7 days was further dichotomised into those that consumed more than 14 units of alcohol the last 7 days or not, in a separate variable.

Levels of physical activity were described by three ordinal variables that examined the frequency that the respondents do rigorous, moderate or mild sports or activities, with possible answers (1) more than once a week, (2) once a week, (3) one to three times a month and (4) hardly ever or never.

## Statistical analysis

Categorical variables are presented as absolute (n) and relative (%) frequencies, while continuous variables are presented using their mean and SD. The Kolmogorov-Smirnov test and normal plots were used to test the normality of the quantitative variable distributions. All the 8529 individuals (of the 9666 initial sample in ELSA wave 7), had usable objective hearing data, measured by a qualified nurse. In total, 257 participants refused to have the assessment (the 2.6% of the full cohort of 9666 participants). As there was no pattern in the missing data regarding age, sex, education, occupation, income and wealth and due to low proportion of missingness (<5%), records with missing data were dropped from the analyses.

We fitted multiple logistic regression models to evaluate the odds of HL at various socioeconomic strata, controlling for gender, age and non-medical determinants of health (BMI, physical activity, tobacco and alcohol consumption). Additionally, we fitted four separate stepwise logistic regression models, to examine the association of HL with non-modifiable (age, gender: step

1), partly modifiable (education, occupation, income, wealth: step 2, respectively) and fully modifiable lifestyle risk factors (BMI, physical activity, tobacco and alcohol consumption: step 3). Age was entered into the multivariable logistic regression models as a continuous variable, to maximise power.

The variants of pseudo R-squared statistics were based on the deviance of the models and used to express how much variance in the outcome is explained by the variables in each stepwise multiple logistic regression model. The variance inflation factor (VIF) was used as an indicator of multicollinearity and the Hosmer-Lemeshow test was used as a post estimation tool, which quantified the goodness-of-fit of the models. For all models, ORs, 95% CIs, unadjusted and adjusted coefficients' beta values, pseudo R$^2$ and mean VIFs are presented. The two-tailed significance level was set ≤0.05. All data were analysed using Stata V.14 (StataCorp, 2015).[23]

## RESULTS

### Sociodemographic characteristics

Overall, 26.6% (2266/8529) of adults aged 50–89 had HL >35 dB HL at 3.0 kHz. The percentages were 32.1% (1198/3728, 95% CI 0.31% to 0.34%) for men and 22.3% (1068/4801, 95% CI 0.21% to 0.23%) for women, respectively. Table 1 shows the distribution of sociodemographic characteristics of the sample (n=8529, aged 50–89) according to hearing acuity. The proportion of men and women with HL >35 dB HL at 3.0 kHz was 52.8 (1198) and 47.2 (1068), respectively. However, men were 1.5 times more likely to have moderately severe or severe HL compared with women. One in three adults aged 65–75 had HL and the percentage of HL in age band 75–89 was threefold larger than in age band 50–64, as one out of every two adults aged 75–89 had HL >35 dB HL at 3.0 kHz.

### Lifestyle factors

Lifestyle factors of the participants are presented in table 2. Over half of the participants were current or former smokers. In addition, patterns of high levels of alcohol consumption among all participants were revealed, with average consumption of more than 14 units of alcohol in the last 7 days for two out of three participants (5223/8528, 95% CI 0.60 to 0.61). Nearly one out of every three of those drinking above the low-risk-level guidelines[22] (1457/5.223, 95% CI 0.27 to 0.29) had HL >35 dB HL at 3.0 kHz.

Three out of four of those with HL >35 dB HL at 3.0 kHz were overweight or obese. Furthermore, those with HL >35 dB HL at 3.0 kHz were twice as likely to hardly ever or never engage in moderate or mild sports activities compared with hearing participants.

### Hearing loss

Table 3 and figure 1 show the results of multiple logistic regression analysis with HL >35 dB HL at 3.0 kHz as the dependent variable and SEP indicators as the independent variables, per each gender. The adjusted odds of HL were higher for those with no qualifications versus those with a degree/higher education (men: OR 1.87, 95% CI 1.47 to 2.38, women: OR 1.53, 95% CI 1.21 to 1.95), those in routine/manual occupations versus those in managerial/professional occupations (men: OR 1.92, 95% CI 1.43 to 2.63, women: OR 1.25, 95% CI 1.03 to 1.54) and those in the lowest versus the highest income and wealth quintiles (men: OR 1.62, 95% CI 1.08 to 2.44, women: OR 1.36, 95% CI 0.85 to 2.16 and men: OR 1.72, 95% CI 1.26 to 2.35, women: OR 1.88, 95% CI 1.37 to 2.58, respectively).

Table 4 shows the summary of stepwise logistic regression analysis for variables predicting HL >35 dB HL at 3.0 kHz. All regression models were statistically significant. Age and gender only explained about 15% of the variance in the likelihood of HL. The addition of lifestyle factors attenuated significantly the association between the HL and SEP indicators and in total the addition of SEP and lifestyle factors in the regression models explained another 10%–15% of the variance in the likelihood of HL. The total variance explained in the overall models containing demographic factors, SEP and lifestyle factors ranged between 25% and 27%. This finding suggests that SEP and lifestyle factors have an equal contribution to HL as age and gender.

The differences in HL prevalence between males and females were observed across all age bands investigated. However, we noticed that the rate of deterioration of hearing acuity as age increases was similar between each age band and nearly to 60% in both genders (figure 2). The difference in prevalence begins at the age band '50–64', where men were twice as likely to have HL.

## DISCUSSION

### Summary of main findings

In this study, we examined whether SEP and modifiable lifestyle factors are associated with HL among older adults in England. We found that variation in education, occupation, income and wealth, which are important determinants of health inequality, are associated with HL. SEP was strongly associated with the likelihood of HL in older adults, with the higher levels of education, income and wealth being less likely to be associated with HL, and the manual occupations increased the likelihood of HL. We also found that socioeconomic and several modifiable lifestyle factors (such as high BMI, physical inactivity, tobacco consumption and alcohol intake above the low-risk-level guidelines[22] are associated with the likelihood of HL as strongly as well-established demographic factors such as age and gender HL. These findings suggest that a large proportion of HL burden is potentially preventable and support the proposition of Scholes et al[6] that there is serious potential to reduce the prevalence and impacts of HL by understanding the impact of socioeconomic inequality in hearing health. Thus, the incidence

**Table 1** Participants sociodemographic characteristics (N=8529, aged 50–89)

| Variable | Normal hearing | HL >35 dB HL at 3.0 kHz | Moderate HL* | Moderately severe or severe HL† |
|---|---|---|---|---|
| Gender | | | | |
| Male | 40.4 (2530) | 52.8 (1198) | 49.5 (741) | 59.5 (457) |
| Female | 59.6 (3733) | 47.2 (1068) | 50.5 (757) | 40.5 (311) |
| Age‡ | 64.3 (9.29) | 69.7 (19.19) | 70.0 (15.85) | 69.1 (24.41) |
| Age group | | | | |
| 50–64 | 51.3 (3135) | 16.2 (349) | 19.3 (280) | 9.8 (69) |
| 65–74 | 34.5 5 (2108) | 33.6 (722) | 36.9 (535) | 26.7 (187) |
| 75–89 | 14.2 (868) | 50.2 (1081) | 43.8 (636) | 63.5 (445) |
| Currently married | | | | |
| No | 31.2 (1908) | 38.4 (826) | 37.5 (544) | 40.2 (282) |
| Yes | 68.8 (4202) | 61.6 (1,326) | 62.5 (907) | 59.8 (701) |
| Retirement status | | | | |
| Retired | 52.4 (3205) | 78.3 (1685) | 76.6 (1112) | 81.3 (573) |
| Not retired | 47.6 (2905) | 21.7 (467) | 23.4 (339) | 18.3 (128) |
| Education | | | | |
| Degree/higher education | 33.7 (1996) | 26.4 (562) | 28.1 (404) | 22.9 (158) |
| A level | 10.0 (596) | 6.4 (137) | 7.0 (100) | 5.4 (37) |
| O level/CSE grade | 24.4 (1448) | 22.3 (473) | 22.4 (321) | 22.0 (152) |
| Foreign/other | 13.5 (798) | 11.9 (252) | 11.9 (171) | 11.7 (81) |
| No qualifications | 18.4 (1090) | 33.0 (701) | 30.6 (439) | 38.0 (262) |
| Occupation based National Statistics Socio-economic Classification | | | | |
| Managerial and professional occupations | 23.4 (1158) | 21.5 (423) | 21.6 (285) | 21.2 (138) |
| Intermediate occupations (non-manual) | 43.4 (2149) | 33.8 (665) | 36.2 (477) | 28.9 (188) |
| Routine and manual occupations | 33.2 (1644) | 44.7 (1643) | 42.2 (1318) | 49.9 (325) |
| Net household income | | | | |
| First quintile (lowest) | 17.0 (872) | 21.3 (421) | 19.7 (262) | 24.8 (159) |
| Second quintile | 18.7 (959) | 24.8 (489) | 24.7 (329) | 24.9 (160) |
| Third quintile | 20.1 (1034) | 23.0 (453) | 22.3 (297) | 24.3 (156) |
| Fourth quintile | 22.5 (1154) | 18.6 (367) | 19.9 (265) | 15.9 (102) |
| Fifth quintile (highest) | 21.7 (1112) | 12.3 (243) | 13.4 (178) | 10.1 (65) |
| Net financial wealth | | | | |
| First quintile (lowest) | 15.5 (794) | 14.7 (290) | 14.9 (199) | 14.2 (91) |
| Second quintile | 17.1 (879) | 24.1 (475) | 22.1 (294) | 28.2 (181) |
| Third quintile | 19.6 (1006) | 23.6 (466) | 23.4 (311) | 24.1 (155) |
| Fourth quintile | 23.5 (1204) | 20.3 (400) | 21.3 (284) | 18.1 (116) |
| Fifth quintile (highest) | 24.3 (1248) | 17.3 (342) | 18.3 (243) | 15.4 (99) |

The header spans: "Hearing acuity % (N) in the better-hearing ear"

Values are expressed as column % (N) unless otherwise is indicated.

*Moderate HL: tones heard at 75 dB HL and 55 dB HL but not at 35 dB HL (the first two of the three tones at 3.0 kHz heard).

†Moderately severe or severe HL: tone heard or not at 75 dB HL and tones not heard at 55 dB HL and 35 dB HL (**0** or 1 of the three tones at 3.0 kHz heard).

‡Mean (SD).

CSE, Certificate of Secondary Education; HL, hearing loss.

**Table 2** Participants' lifestyle factors (N=8529, aged 50–89)

| Variable | Hearing acuity % (N) in the better-hearing ear | | | |
| --- | --- | --- | --- | --- |
| | Normal hearing | HL>35 dB HL at 3.0 kHz | Moderate HL* | Moderately severe or severe HL† |
| **Tobacco consumption (any type of nicotine products)** | | | | |
| Current | 11.7 (712) | 10.0 (215) | 9.6 (139) | 10.8 (76) |
| Former | 49.0 (2996) | 56.7 (1219) | 55.8 (810) | 58.4 (409) |
| No of cigarettes smoked per day‡ | 12.79 (14) | 12.79 (13) | 12.69 (13) | 11.90 (12) |
| Never | 39.3 (2403) | 33.3 (718) | 34.6 (502) | 30.8 (216) |
| **Alcohol consumption (in the last 7 days)** | | | | |
| No of days of alcohol consumption§ | 3 (3) | 3 (4) | 3 (4) | 3 (4) |
| No of measures of spirit‡ | 2.1 (2) | 2.3 (3) | 2.2 (3) | 2.6 (3) |
| No of glasses of wine‡ | 4.3 (6) | 3.6 (5) | 3.9 (6) | 3.1 (4) |
| No of pints of beer‡ | 2.1 (2) | 2.3 (3) | 2.3 (3) | 2.4 (3) |
| Total units of alcohol in the last 7 days‡ | 15.0 (18) | 14.2 (19) | 14.5 (21) | 13.5 (17) |
| Consumption of more than 14 units | 61.6 (3766) | 67.7 (1457) | 67.3 (977) | 68.5 (480) |
| **BMI Classification** | | | | |
| Underweight | 3.4 (160) | 5.0 (92) | 4.9 (60) | 5.3 (32) |
| Normal | 26.9 (1255) | 20.6 (376) | 19.6 (239) | 22.7 (137) |
| Overweight | 40.0 (1869) | 42.8 (780) | 41.4 (506) | 45.4 (274) |
| Obese | 29.7 (1390) | 31.6 (576) | 34.1 (416) | 26.6 (160) |
| **Physical activity** | | | | |
| **Frequency does rigorous sports or activities** | | | | |
| More than once a week | 23.0 (1407) | 14.3 (307) | 16.1 (233) | 10.6 (74) |
| Once a week | 10.3 (626) | 7.0 (151) | 7.9 (115) | 5.1 (36) |
| One to three times a month | 10.1 (617) | 7.1 (153) | 7.6 (111) | 6.0 (42) |
| Hardly ever or never | 56.6 (3459) | 71.6 (1541) | 68.4 (992) | 78.3 (549) |
| More than once a week | 68.4 (4180) | 51.3 (1104) | 53.7 (780) | 46.2 (324) |
| Once a week | 12.6 (771) | 13.6 (292) | 14.1 (204) | 12.6 (88) |
| One to three times a month | 5.9 (360) | 7.8 (169) | 7.6 (110) | 8.4 (59) |
| Hardly ever or never | 13.1 (799) | 27.3 (587) | 24.6 (357) | 32.8 (230) |
| **Frequency does mild sports or activities** | | | | |
| More than once a week | 83.9 | 73.7 | 76.0 (1103) | 68.9 (483) |
| Once a week | 8.2 | 10.1 | 9.8 (142) | 10.5 (74) |
| One to three times a month | 2.3 | 3.5 | 3.3 (48) | 4.0 (28) |
| Hardly ever, or never | 5.6 | 12.7 | 10.9 (158) | 16.6 (116) |

Values are expressed as column % (N) unless otherwise is indicated.
*Moderate HL: tones heard at 75 dB HL and 55 dB HL but not at 35 dB HL (the first two of the three tones at 3.0 kHz heard).
†Moderately severe or severe HL: tone heard or not at 75 dB HL and tones not heard at 55 dB HL and 35 dB HL (0 or 1 of the three tones at 3.0 kHz heard).
‡Mean (SD).
§Median (Range).
BMI, body mass index; HL, hearing loss.

and severity of HL in England could be significantly reduced by the governmental policies to mitigate socio-economic disparities and public health interventions to promote healthier lifestyles in middle-aged and older adults in England. The occurrence of objective hearing data eliminated the different types of bias that occur in self-reporting hearing difficulties,[24] strengthening the accuracy of findings.

### Strengths and limitations
The main strength of our study was that is the first to examine the association of four separate SEP indicators

**Table 3** Multiple logistic regression analysis of n=8529, aged 50–89 with HL >35 dB HL at 3.0 kHz in better-hearing ear as dependent variable and SEP indicators as independent variables

| | Unadjusted OR (95% CI)* | | Adjusted OR (95% CI)† | |
| --- | --- | --- | --- | --- |
| | **Men** | **Women** | **Men** | **Women** |
| **Education** | | | | |
| No qualifications | 2.39 (1.96 to 2.90) | 2.67 (2.20 to 3.24) | 1.87 (1.47 to 2.38) | 1.53 (1.21 to 1.95) |
| Foreign/other | 1.06 (0.83 to 1.36) | 1.37 (1.07 to 1.74) | 1.46 (1.09 to 1.94) | 0.99 (0.74 to 1.32) |
| O level/CSE grade | 1.56 (1.29 to 1.89) | 1.00 (0.80 to 1.25) | 1.42 (1.13 to 1.79) | 0.94 (0.73 to 1.22) |
| A level | 1.01 (0.77 to 1.32) | 0.69 (0.50 to 0.97) | 1.08 (0.78 to 1.51) | 0.82 (0.56 to 1.21) |
| Degree/higher education (reference) | | | | |
| **Occupation based National Statistics Socioeconomic Classification** | | | | |
| Routine and manual occupations | 1.69 (1.39 to 2.08) | 1.35 (1.15 to 1.59) | 1.92 (1.43 to 2.63) | 1.25 (1.03 to 1.54) |
| Intermediate occupations (non-manual) | 1.47 (1.23 to 1.75) | 1.54 (1.19 to 1.96) | 1.61 (1.25 to 2.08) | 1.35 (1.01 to1.85) |
| Managerial and professional occupations (reference) | | | | |
| **Net household income** | | | | |
| First quintile (lowest) | 1.94 (1.50 to 2.52) | 3.04 (2.31 to 3.99) | 1.62 (1.08 to 2.44) | 1.36 (0.85 to 2.16) |
| Second quintile | 2.12 (1.67 to 2.70) | 3.00 (2.28 to 3.93) | 1.31 (0.93 to 1.85) | 1.40 (0.89 to 2.18) |
| Third quintile | 1.98 (1.56 to 2.51) | 2.31 (1.75 to 3.05) | 1.40 (1.01 to 1.94) | 1.08 (0.69 to 1.67) |
| Fourth quintile | 1.38 (1.08 to 1.74) | 1.65 (1.23 to 2.20) | 1.09 (0.80 to 1.49) | 1.08 (0.70 to 1.66) |
| Fifth quintile (highest) (reference) | | | | |
| **Net financial wealth** | | | | |
| First quintile (lowest) | 1.11 (0.86 to 1.45) | 1.79 (1.38 to 2.33) | 1.72 (1.26 to 2.35) | 1.88 (1.37 tro 2.58) |
| Second quintile | 1.92 (1.52 to 2.42) | 2.39 (1.88 to 3.04) | 1.66 (1.26 to 2.18) | 1.33 (1.00 to 1.77) |
| Third quintile | 1.63 (1.30 to 2.04) | 1.95 (1.53 to 2.50) | 1.45 (1.12 to 1.88) | 1.41 (1.06 to 1.88) |
| Fourth quintile | 1.06 (0.85 to 1.32) | 1.48 (1.15 to 1.91) | 0.96 (0.75 to 1.24) | 1.26 (0.94 to 1.68) |
| Fifth quintile (highest) (reference) | | | | |

*Unadjusted OR
†OR adjusted for age, marital status, retirement status, body mass index, tobacco consumption, alcohol consumption and physical activity.
CSE, Certificate of Secondary Education; HL, hearing loss; SEP, socioeconomic position.

with HL among older adults in England, instead of a proxy measure to reflect one's total SEP, capturing, therefore, most of the variation in socioeconomic stratification[12] and also the role of modifiable lifestyle risk factors in these associations. Another strength is that the analyses were based on a representative cohort of 8529 participants contained in ELSA, which is a rich resource of information on the dynamics of health, social, well-being and economic circumstances in the English population aged 50 and older.[16]

However, there are also important limitations. First, no causal or temporal relationships can be established between lifestyle factors and HL in this cross-sectional study. Unhealthy lifestyle behaviours could lead to HL in older people but is also possible that older people adopt less healthy lifestyles after HL. Second, all the analysed factors explained less than one-third of the variance for the prevalence of HL suggesting that there are additional major factors associated with HL in older adults which have not been included in our analyses. Longitudinal analyses using a broader range of physical health, mental

health and social care variables are highly recommended to obtain a comprehensive understanding of modifiable factors which contribute to HL among older adults in England. Third, the ELSA dataset did not include information concerning the occupational and social noise exposure, which has a damaging effect in hearing.[4] We, therefore, were not able to examine the association of noise exposure with smoking in the relationship of SEP with HL, as in a previous study which found that the smoking habit in workers exposed to occupational noise greatly influenced HL.[25] However, we examined the association of manual occupations with HL and its attenuation by modifiable determinants including smoking habit, which is of a higher prevalence among those that work in routine and manual occupations in England.[13] Finally, we did not run weighted analyses which may have reduced the generalisability of our findings, as the ELSA sample members at wave 7 could be healthier on average than the population, potentially resulting in an underestimation of relationships.

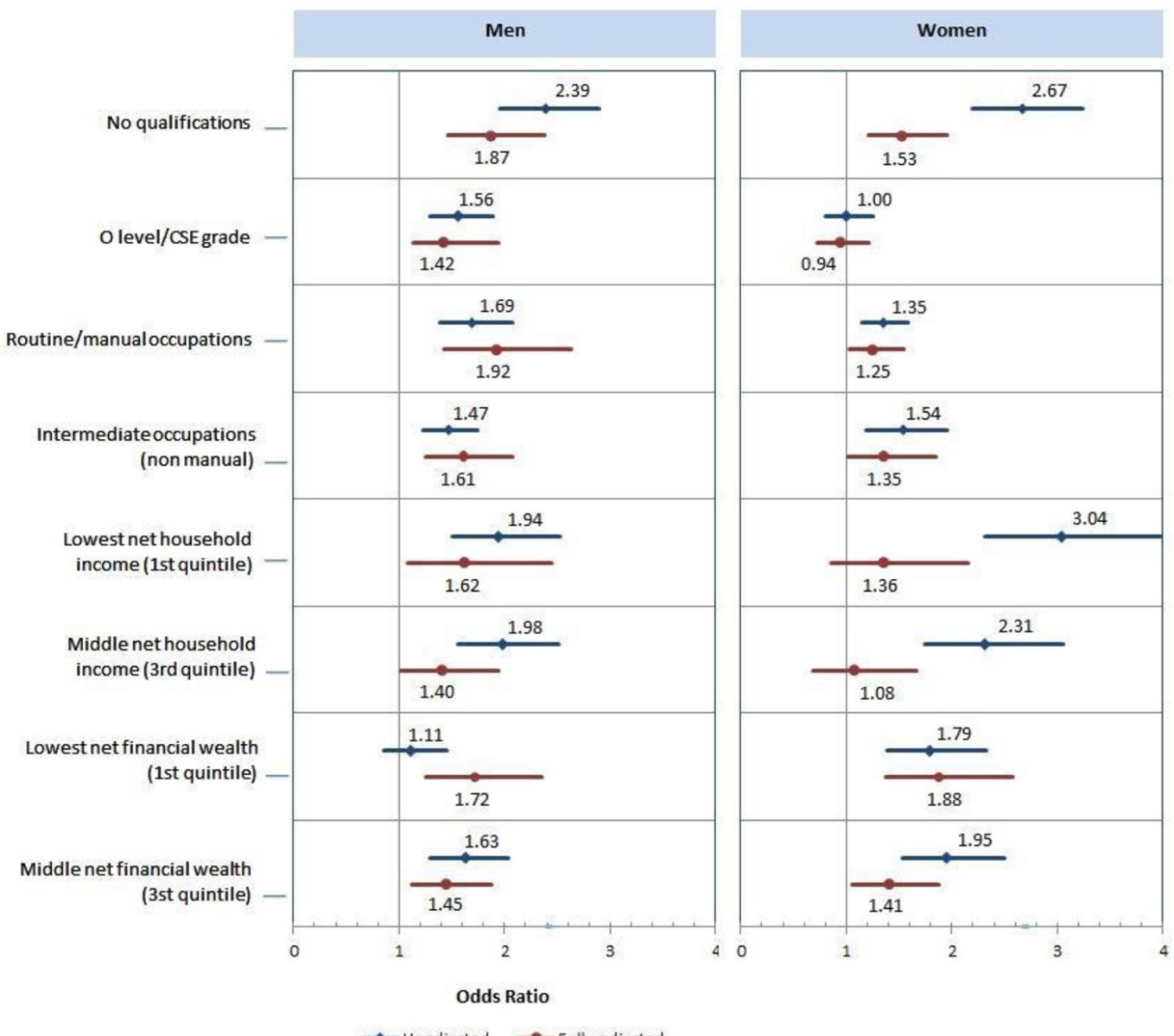

**Figure 1** Associations between socioeconomic position and hearing loss in middle-aged and older adults (n=8529, aged 50–89). Indicators of SEP were categories of the highest educational attainment (degree/higher education as a reference), tertiles of self-reported occupation based on the National Statistics Socio-economic Classification (managerial and professional as reference), quintiles of the net household income (first quintile lowest; fifth quintile highest) and quintiles of the total non-pension wealth that is reported at the household level (first quintile lowest; fifth quintile highest). lines represent or (outcome=hearing loss) and its 95% CI. Model A (rhombus): unadjusted. model B (circles): adjusted for age, marital status, retirement status, body mass index, tobacco consumption, alcohol consumption and physical activity. CSE, Certificate of Secondary Education; SEP, socioeconomic position.

## Research and policy implications

A number of previous studies have reported that the odds of HL in older adults were significantly increased for those with lower educational attainment.[6 10 26 27] VS and those in manual versus non-manual occupations,[28–31] Besides, income is a correlate of HL, with the prevalence of untreated HL being higher among low-income older adults in the USA.[31] In our study, those in the lowest quintile of net household income had disproportionally higher percentages of moderate HL compared with moderately severe or severe HL, but this pattern was not found in the quintiles of wealth, as expected. This may indicate a possible delay in diagnosis of hearing problems among those in lower SEP due to financial barriers in access to health services,[32] which needs further exploration, as HL is highly undiagnosed and untreated among older adults in England.[20]

International studies have also shown that tobacco consumption, high body mass and high fat and high calorie food consumption can have an adverse impact on hearing,[11 33–35] On the other hand, a higher level of physical activity is related with a lower risk of HL.[34] In our

**Table 4** Summary of stepwise logistic regression coefficients for variables predicting HL >35 dB HL at 3.0 kHz in the better-hearing ear (n=8529, aged 50–89), according to different SEP indicators (education, occupation, income and wealth)

| Step/predictor | Model A | | | Model B | | | Model C | | | Model D | | |
|---|---|---|---|---|---|---|---|---|---|---|---|---|
| | Step 1 | Step 2a | Step 3 | Step 1 | Step 2b | Step 3 | Step 1 | Step 2 c | Step 3 | Step 1 | Step 2d | Step 3 |
| 1. Non-modifiable | (Education) | | | (Occupation) | | | (Income) | | | (Wealth) | | |
| Gender (female) | −0.62*** | −0.59*** | −0.72*** | −0.62*** | −0.64*** | −0.68*** | −0.62*** | −0.69*** | −0.70*** | −0.62*** | −0.69*** | −0.62*** |
| Age | .12*** | .11*** | .10*** | .12*** | .13*** | .11*** | .12*** | .11*** | .11*** | .12*** | .11*** | .12*** |
| 2. Partly modifiable | | | | | | | | | | | | |
| 2a. Education | | −0.15*** | −0.11*** | | – | – | | – | – | | – | – |
| 2b. Occupation (manual) | | | – | | .26*** | .20*** | | – | – | | – | – |
| 2c. Net Household Income | | | – | | – | – | | −0.14*** | −0.09*** | | – | – |
| 2d. Net Financial Wealth | | | – | | – | – | | – | – | | −0.17*** | −0.11*** |
| 3. Modifiable | | | | | | | | | | | | |
| Smoking (current/former) | | | .10* | | | 0.09 | | | .10* | | | 0.09** |
| Alcohol consumption (>14 units per week) | | | .24*** | | | .19*** | | | .17*** | | | 0.18** |
| Body mass index (<25) | | | −0.05* | | | −0.06 | | | −0.03 | | | −0.04 |
| Physical activity (rigorous sports or activities, once or more/week) | | | −0.14*** | | | −0.16*** | | | −0.12*** | | | −0.13*** |
| Physical activity (moderate sports or activities, once or more/week) | | | −0.24*** | | | −0.24*** | | | −0.24*** | | | −0.24*** |
| Physical activity (mild sports or activities, once or more/week) | | | −0.17*** | | | −0.15*** | | | −0.15*** | | | −0.14*** |
| Pseudo R² | 0.15 | 0.18 | 0.28 | 0.15 | 0.19 | 0.26 | 0.17 | 0.18 | 0.29 | 0.17 | 0.18 | 0.27 |
| Δ Pseudo R² | – | 0.03 | 0.10 | – | 0.04 | 0.07 | – | 0.01 | 0.11 | – | 0.01 | 0.09 |
| Mean VIF | – | – | 1.16 | – | – | 1.20 | – | – | 1.24 | – | – | 1.15 |

*P<0.05, **P<0.01, ***P<0.001.
HL, hearing loss; SEP, socioeconomic position; VIF, variance inflation factor.

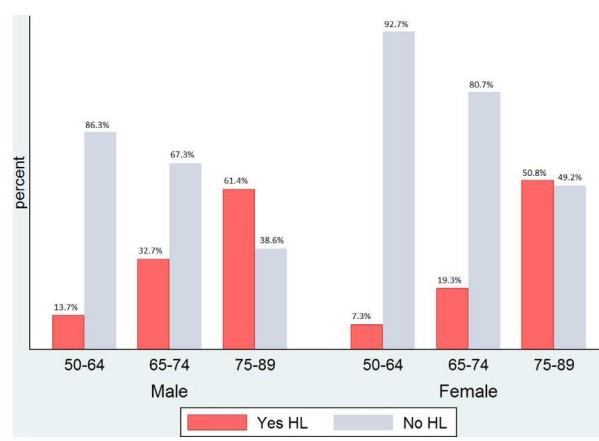

**Figure 2** Hearing loss (HL) by age group and gender* (n=8529 participants, aged 50–89, from the seventh wave of the English Longitudinal Study of Ageing. HL was defined as >35 dB HL at 3.0 kHz, in the better-hearing ear. *Prevalence estimates for males (N=3728) and females (N=4801).

study, two out of three participants were drinking more than the low-risk level of the 14 units of alcohol a week.[22] We considered, therefore, that alcohol consumption above the low-risk-level guidelines may play an important role in the association between SEP and HL among the English population and thus we included this variable in the regression models, which has not been previously examined in the literature for the English population. Our findings showed that drinking above the low-risk-level guidelines increased the likelihood of HL. This finding is in line with Chief Medical Officer's Drinking guidelines,[22] which suggest that it is safest not to drink regularly more than 14 units per week, to keep health risks from drinking alcohol to a low level.

The associations between indicators of lower SEP and HL may be markers of less healthy lifestyle,[5] which may explain the link between HL and socioeconomic and lifestyle factors investigated. Cruickshanks *et al*[36] did not find significant associations between hearing impairment and BMI, smoking and alcohol in the multivariable analyses using a younger population-based sample (aged 18–74 years) of Hispanics/Latinos. Hence, it is likely that HL in older population (eg, 50 years and above) is associated with different risk factors or combinations of socioeconomic and lifestyle risk factors across the life course.

The higher prevalence of HL among men aged 50 and above compared with women has also been reported in other studies.[3 6] However, we observed that the rate of deterioration of hearing acuity as age increases was similar between each age band and nearly to 60% in both genders. The difference in prevalence begins at the age band '50–64', where men were twice as likely to have HL. Thus, the differences in modifiable lifestyle factors that were revealed in the stepwise regression models may finally explain why the male sex is often cited as consistent risk factor for HL,[35–37] leading to the exploration of modifiable determinants that are common in both

genders[5] and paving the way for interventions to improve the population's hearing health.

In terms of policy, generating evidence concerning the critical variables associated with HL is an important step in designing targeted services and interventions for individuals that face hearing health inequalities, and especially, for those in the lowest SEP groups, where the burden of HL falls highest. This is of major importance for the population in England, as sensor diseases are the first leading cause of morbidity among adults 70 years and older and the second leading cause among adults 50–69 years.[13] Our findings support the view that HL is a non-communicable disease,[38] which can be prevented or ameliorated by the governmental policies to mitigate socioeconomic disparities and public health interventions to promote healthier lifestyles in middle-aged and older adults in England.

## CONCLUSION

The main finding of our study is that HL is strongly associated with socioeconomic factors and modifiable lifestyle behaviours. Our findings are supportive of a new conceptualisation of HL which argues that HL is not necessarily an inevitable accompaniment of ageing, but also a potential preventable lifestyle disease, paving the way for the term lifestyle-related HL, where lifestyle refers to social practices and ways of living adopted by individuals that reflect personal, group and socioeconomic identities,[39] instead of the non-inclusive term 'age-related HL'. Future research in hearing health inequalities should investigate the role of the prolonged exposure to these modifiable lifestyle behaviours in the development of HL and the role of other comorbid chronic diseases in the elderly.

**Author affiliations**
[1]Centre for Primary Care and Health Services Research, Division of Population Health, School of Health Sciences, Faculty of Biology, Medicine and Health, University of Manchester, Manchester, UK
[2]Division of Informatics, Imaging & Data Sciences, School of Health Sciences, Faculty of Biology, Medicine and Health, University of Manchester, Manchester, UK
[3]NIHR Greater Manchester Patient Safety Translational Research Centre, Division of Pharmacy and Optometry, School of Health Sciences, Faculty of Biology, Medicine and Health, University of Manchester, Manchester, UK
[4]NIHR Greater Manchester Patient Safety Translational Research Centre, Division of Population Health, School of Health Sciences, Faculty of Biology, Medicine and Health, University of Manchester, Manchester, UK

**Acknowledgements** DT would like to acknowledge the contribution of Dr Piers Dawes and Prof Neil Pendleton, who had provided assistance in obtaining funding and were the former supervisors for her NIHR Manchester Biomedical Research Centre PhD Studentship, but did not fulfil the criteria set out in the Authorship Guidelines of The University of Manchester to be listed as authors.

**Contributors** DT, EK, DA and MP were responsible for developing the design of the study. DT was responsible for conducting the analyses, interpreting the results and drafting the manuscript. DT, EK, DA and MP critically revised the manuscript. All authors have read and approved the final manuscript.

**Funding** This research was funded by the NIHR Manchester Biomedical Research Centre (BRC).

**Disclaimer** The views expressed are those of the authors and not necessarily those of the BRC, the NIHR or the Department of Health.

**Competing interests**  No, there are no competing interests for any author.

**Patient consent for publication**  Not required.

**Provenance and peer review**  Not commissioned; externally peer reviewed.

**Data availability statement**  The English Longitudinal Study of Ageing dataset is available via the UK Data Service (http://www.ukdataservice.ac.uk). Statistical code is available from the corresponding author at dialechti.tsimpida@manchester.ac.uk.

**Open access**  This is an open access article distributed in accordance with the Creative Commons Attribution 4.0 Unported (CC BY 4.0) license, which permits others to copy, redistribute, remix, transform and build upon this work for any purpose, provided the original work is properly cited, a link to the licence is given, and indication of whether changes were made. See: https://creativecommons.org/licenses/by/4.0/.

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
