## [Reviewer comments · BMJ Open]

ARTICLE DETAILS

TITLE (PROVISIONAL)	Socioeconomic and Lifestyle Factors Associated with Hearing Loss in Older Adults: A Cross-sectional Study of the English Longitudinal Study of Ageing (ELSA)
AUTHORS	Tsimpida, Dialehti; Kontopantelis, Evangelos; Ashcroft, Darren; Panagioti, Maria

VERSION 1 – REVIEW

REVIEWER	Shaun Scholes University College London, United Kingdom
REVIEW RETURNED	28-May-2019

GENERAL COMMENTS	Using data collected from the seventh wave of the English Longitudinal Study of Ageing (ELSA), the authors examined the associations between socioeconomic position (SEP) and objectively measured hearing loss (HL: defined as >35dB at 3.0 kHz), and between modifiable lifestyle factors (e.g. high BMI and physical inactivity) and HL. The authors found SEP and modifiable lifestyle factors to be strongly associated with HL. I was the first author of a similar study using Health Survey for England data published in BMJ Open (using the same objective test as the present study). As with a previous review for BMJ Open, I stress to the Editors and authors that I am not an expert on hearing (and so cannot comment on very technical matters). This is a reasonably well written study (but needs careful proof reading), but I do have a number of concerns (including statements that the data is representative and the lack of clarity of the results and rationale for Table 4). I have a number of comments the authors may wish to consider in a revised version. Abstract The metric of HL should be dB HL (as in the first paragraph of the introduction) not dB. In the conclusion, the authors mention strategies to improve wellbeing of the older population. Yet wellbeing was not analysed in this study. The authors should instead make a final statement about HL. Strengths and limitations Please critically evaluate use of the term “excessive smoking”. The term excessive drinkers is also used in the results: is this an appropriate term? Introduction
--

	The last sentence needs revising. Firstly, ELSA is not “the English adult population”. Secondly, perhaps this study quantifies the variance in HL: but cannot say “how” (unless the authors mean in a strictly statistical sense). I think the Introduction lacks evidence on the associations between the four lifestyle risk factors considered and HL. Table 2 for example contains many indicators of alcohol consumption: are these especially relevant for studying HL, if so, it should be briefly referred to in the Introduction. Methods: study population It would be easier for readers if you cited the ELSA Cohort profile (published in The International Journal of Epidemiology). Outcomes: hearing loss Lines 41-42: It is the research team who define the severity of HL, not the HearCheck device? Variables The authors should specify when (i.e. at which wave) the variables were assessed. Statistical analysis The authors’ statement on missing data lacks detail. I find it surprising that “there were no missing values in the hearing data of the final analytical sample”. Can the authors check whether some participants gave consent, did not have an ear infection, but for some reason had no usable hearing data? Furthermore, what were the differences (if any) between the participants at Wave 7 and those in the analytical sample? The amount of missing data may have been low (<5%) but were there any systematic differences between participants with and without missing data? Does this effect the representativeness of the findings? Furthermore, the authors make no mention of using any weights to increase representativeness of the ELSA data. As an ageing cohort study, it is likely that the ELSA sample used in this study (without weights) is not representative but is likely to be healthier and wealthier than the older household English population. The potential bias I mentioned in the HSE study will in all likelihood be stronger with ELSA data. Finally, there is no information in the statistical analysis section to explain which measure indicates how much variance in the outcome is explained by the variables in the model (as outlined in the strengths and limitations of the study box). Presumably, this is the Pseudo R2 but this needs to be clarified for the reader. Results The 95% Confidence Intervals should be in %s (not 0.306 to 0.337). The estimates in Table 1 should have one decimal point only (two d.p. is spurious accuracy). Table 1 should contain an explanation of how moderate and severe HL was defined. The description of the results in Table 2 is confusing: I recommend that the authors focus on the % of HL across the subgroups (as set out in the Table: hearing acuity (%)), rather than the % obese say
--	---

	among those with HL. Can the authors check whether they mean SD rather than SE of the mean (Table 2). I am confused about the differences between Table 3 and Table 4. Is it necessary to present both these analyses? Why do the authors show odds ratios in Table 3, but logistic coefficients in Table 4? Why is Table 3 sex-specific but Table 4 not so? At first glance, this is confusing. The write up of Table 4 is also very short. Adding lifestyle factors is stated to increase the variance explained: but there is no commentary on the associations between lifestyle risk factors and HL. The results in Table 4 are not immediately clear (e.g. reference categories not given). For example, why are the coefficients for education negative but the coefficients for occupation positive? It would help if you coded them all the same way (e.g. comparing lower SEP to higher SEP on all four measures: as you do in Table 3). Discussion The authors should not claim to show any evidence of the effects of SEP on HL (first sentence of the Strengths and Limitations section). As mentioned above, the authors have too little to say on the associations between RF and HL. In the Discussion, the authors explain why they looked at alcohol: but then do not explain what they found and what the implications are. Other comments The authors use England and UK interchangeably; ELSA is an English ageing cohort.
--	--

REVIEWER	Lei Yang Hangzhou Normal University. Hangzhou, China.
REVIEW RETURNED	10-Jun-2019

GENERAL COMMENTS	In this study, the author investigated the association between socioeconomic and lifestyle factors with hearing loss in older adults, using a cross-sectional study. The notion that socioeconomic and lifestyle factors are associated with HL among older adults as strongly as core demographic risk factors is interesting. In general, the manuscript is simple and detailed, the thinking is clear. The design is reasonable, the sample size is relatively large, the indicators are described in detail, the statistical methods are simple, and the discussion is profound, which can provide suggestions for the local government to formulate health policy strategies. However, there are some minor issues that the authors should answer to improve the quality of the manuscript.  1. The definition of hearing loss is >35dB at 3kHz in the better hearing ear. Why 35dB and 3kHz? As far as I know, 8kHz is closely related to hearing loss in the elderly. The authors mentioned that audiometry is measured at 1kHz and 3kHz. Why? 2. The diabetes, hypertension, coronary heart disease and other chronic diseases among the elderly will have an impact on your results. Have you considered this part? 3. I have noticed the forest graph corresponding to table 3. Why does the author choose only some of the information in each socioeconomic indicator?
---

	4. The author mentioned that " This study is the first that examines the effects of four different SEP indicators (education, occupation, income, wealth) to the objectively measured HL in older adults." Can you explain how these four socioeconomic indicators are selected? And other similar studies seem to have looked at these indicators. Has the author considered synthesizing them into a new socioeconomic indicator? It's probably something no one else has done.
--	---

REVIEWER	Adele Goman Johns Hopkins University, USA
REVIEW RETURNED	14-Jun-2019

GENERAL COMMENTS	This manuscript aims to examine the relationship between socioeconomic factors and hearing loss in adults aged 50-89. Additionally the manuscript aims to examine whether lifestyle factors are associated with hearing loss. To achieve these aims the authors report cross-sectional analyses using data from wave 7 of the English Longitudinal Study of Ageing. The research question addresses a gap in the literature as research on the relationship between socioeconomic and lifestyle factors and hearing loss is limited. A strength of the study is the use of a representative sample of adults aged 50 and older. I have some concerns with the analysis and conclusions drawn and have outlined these below. As a general comment, as this paper aims to examine the relationship between hearing loss and socioeconomic/lifestyle factors, the manuscript would have benefited from some discussion on the potential links between hearing loss and the socioeconomic and lifestyle factors investigated. Gold standard pure-tone-audiometry was not available in this study but it would be helpful if the authors could include a bit of background on how the screening device used compares with standard definitions of hearing loss through audiometry. The phrasing of the definition of moderately severe or severe HL is somewhat confusing and may benefit from dropping "(0 or 1 of the three tones at 3.0 kHz heard)" Page 6: I would recommend the authors to separate out the definition of normal hearing from the definition of moderately severe or severe HL into a third "c" category for clarity. The definition of occupation in the study could be clearer. The authors state "Tertiles of self-reported occupation were based on the National Statistics socio-economic classification (NS-SEC): managerial and professional; intermediate; routine and manual occupations." which suggests occupation was defined on the level of the participant. However the authors then go on to state "This was based on the household reference person with the highest income" which suggests it was based on the household level. Please could the authors clarify whether the classification of occupation was on the individual or household level. If, on the household level, please would the authors justify why the occupation of the person with the highest income was used rather than the participant, given the proposed links between manual occupation and noise exposure raised by the authors.
--

	The author's state: "Thus, it was possible that the household reference person was a woman" which is evident from the previous statement and therefore could be dropped. Please would the authors indicate if age was treated as a continuous or categorical variable in the regression models. Tables 1 and 2 are presented with percentages calculated row-wise which is less conducive to the reader's understanding than percentages calculated column wise would be. I.e. The authors indicate that 32.14% of males and 22.25% of females have HL >35dB at 3.0 kHz. What may be more informative would be to know the proportion of individuals with HL >35dB who are male and female. It would be useful if the authors could add "better-ear" when describing hearing acuity in tables 1 and 2. The footnotes of tables 1 and 2 state "Values are expressed as n (%)" but the reverse "% (n)" is actually shown in the tables. This should be corrected. The analyses and results shown in Table 4 need more explanation. Why has age (mean) been used rather than age as a continuous variable? Why is only one category of each of the lifestyle variables shown? Showing the crude and adjusted odds of each variable on hearing loss would be informative here. The discussion introduces figure 2 which would be better introduced in the results section. Additionally the discussion states that the difference in prevalence "begins" at 50-64 whereas that was the earliest age band included in these analyses. Therefore a more accurate description would be to state that differences in hearing loss prevalence between males and females were observed across all age bands investigated. The authors state that "male sex is not a consistent risk factor in studies" and cite Cruickshanks et al 1998 as a reference for this statement. However, in the Cruickshanks et al study hearing loss was greater for men than women and remained so after adjustment for demographic factors, noise exposure, and occupation. Male sex is often cited as a consistent risk factor for hearing loss (e.g. Hoffman et al, 2016; Cruickshanks et al, 2015; Lin et al, 2011) The authors state in the abstract and discussion that "socioeconomic and lifestyle factors" (abstract) and "several modifiable lifestyle factors" (discussion) are associated with hearing loss "as strongly" as the demographic risk factors of age and sex. However, I do not think such a conclusion is justified based on the results presented. Table 4 illustrates a small increase in the variance explained by socioeconomic factors compared to age and sex only. When several lifestyle factors are added in the model the variance explained does increase but the collective effect of all the lifestyle factors examined is still less than that of the demographic risk factors. The authors should discuss how their findings compare with Cruickshanks et al, 2015 who also investigated the impact of occupation and some lifestyle factors (BMI, alcohol, smoking) on hearing loss.
--	---

REVIEWER	Sten Hellström CLINTEC, Div of ENT, Karolinska Institutet, Stockholm, Sweden
REVIEW RETURNED	15-Jun-2019

GENERAL COMMENTS	The present manuscript describes an exciting study on possible associations between socioeconomic and lifestyle factors and HL among older adults in England. The study population, n= 8,529 participants, was available from an ongoing prospective English longitudinal study of aging focussing on social, wellbeing and economic circumstances. The study shows that socioeconomic and modifiable lifestyle factors are associated with HL among older adults as strongly as core demographic risk factors, such as age and gender. However, the analysed factors explained less than 30% of the variance for the occurrence of HL, which means that there are additional major factors associated with HL in older adults, which still has to be investigated and identified. One such important factor, which was not registered in the study population was possible influence of exposure to noise. Despite these shortcomings this study support the suggestions that HL may be prevented or improved by public health policies and interventions. The study is well designed and well performed and the results original. I am ready to recommend this manuscript for publication in BMJ Open., but before that I would like to have the authors to answer the comments and questions below:  - “older adults” in this study are aged 50 to 89 years of age. Which is the background to this definition of older adults and not e.g. 65 to 90? In Table I the HL is shown for 3 age groups; 50-64, 65-74 and 75-89. I should have been of interest to have these different age groups for most variables, not the least concerning the modifiable lifestyle factors. Similarly I have difficulties to see the relevance of Retirement status in Table 1 if not correlated to age groups with in the age span of 50-89. - The objective measurement of hearing acuity was performed by a handheld screening device, HearCheck. In p 6 it is described that the hearing test is performed with a series of three sounds at 1.0 kHz and 3.0 kHz. For the determination of the hearing loss only measurements at 3.0kHz are used and described in text. Why including 1.0 kHz in the test situation and then not using it to characterize the HL? - The HearCheck has been used in some other scientific studies. How reliable is the present method for determining the degree of hearing loss? - p 7 – When it comes to categories of some variables there are difficulties to convert e.g educational levels to those of other countries. How should e.g. A level and O levels CSE be translated to a more international language? And what stands CSE for? - p 9 – last sentence – HL in various age groups is described and has been commented upon above. However, though only used in relation to HL the subdividing in groups should have been mentioned in the Methodology section.
--

VERSION 1 – AUTHOR RESPONSE

Reviewer: 1

1. Using data collected from the seventh wave of the English Longitudinal Study of Ageing (ELSA), the authors examined the associations between socioeconomic position (SEP) and

objectively measured hearing loss (HL: defined as >35dB at 3.0 kHz), and between modifiable lifestyle factors (e.g. high BMI and physical inactivity) and HL. The authors found SEP and modifiable lifestyle factors to be strongly associated with HL. I was the first author of a similar study using Health Survey for England data published in BMJ Open (using the same objective test as the present study). As with a previous review for BMJ Open, I stress to the Editors and authors that I am not an expert on hearing (and so cannot comment on very technical matters). This is a reasonably well written study (but needs careful proof reading), but I do have a number of concerns (including statements that the data is representative and the lack of clarity of the results and rationale for Table 4). I have a number of comments the authors may wish to consider in a revised version.

Response: We would like to thank the reviewer for a thorough review and detailed recommendations, which substantially improved our manuscript. We have now revised the text to make clear that ELSA study is representative of the English population aged 50 years old and above and we provided a detailed response in comment 17, regarding the coefficients in Table 4.

Changes in the manuscript: We revised the text accordingly (page 3, line 11; page 13, lines 5, 27, 31, 44, page 14, line 22).

2. Abstract: The metric of HL should be dB HL (as in the first paragraph of the introduction) not dB

Response: This was a typo; we have revised it throughout the text.

3. In the conclusion, the authors mention strategies to improve wellbeing of the older population. Yet wellbeing was not analysed in this study. The authors should instead make a final statement about HL.

Response: We thank the reviewer for this comment. We now mention that providing evidence concerning the critical variables associated with HL is an important step in designing targeted services and interventions for individuals that face hearing health inequalities especially of those in the lowest SEP groups, where the burden of HL fall highest.

Changes in the manuscript: We have now revised the sentences on page 2 (line 55) and page 16 (line 16) and deleted the text regarding strategies to improve wellbeing of the older population.

4. Please critically evaluate use of the term “excessive smoking”. The term excessive drinkers is also used in the results: is this an appropriate term?

Response: We thank the reviewer for his comment. We follow the Chief Medical Officers’ (CMO) low risk drinking guidelines, which for both men and women is that “to keep health risks from alcohol to low level it is safest not to drink more than 14 units a week on a regular basis”. We now do not mention the term “excessive drinkers” so as to avoid misconceptions regarding the frequency and volume of heavy drinking (e.g. in case of not spreading drinking occasions over multiple days if weekly consumption is below the 14 unit limit). We also amended the term “excessive smoking” to “smokers (current/former), according to the classification of tobacco consumption on page 8, lines 9-14.

Changes in the manuscript: We changed the term “excessive drinkers” to “alcohol intake above the low risk level guidelines” and the term “excessive smokers” to “smokers (current/former)” throughout the manuscript.

5. Introduction: The last sentence needs revising. Firstly, ELSA is not “the English adult population”.

Response: Thank you for your careful reading. This was a typo, we have revised the text as “the English population aged 50 years old and above”.

Changes in the manuscript: We revised the sentences on page 5 (line 14).

6. Perhaps this study quantifies the variance in HL: but cannot say “how” (unless the authors mean in a strictly statistical sense). I think the Introduction lacks evidence on the associations between the four lifestyle risk factors considered and HL. Table 2 for example contains many indicators of alcohol consumption: are these especially relevant for studying HL, if so, it should be briefly referred to in the Introduction.

Response: We thank the reviewer for his helpful comment that improved the clarity of the introduction. We now mention that there is evidence that several modifiable lifestyle factors, such as smoking, alcohol consumption, high body mass index and physical inactivity are associated with hearing health. Of course, the causal paths are not clear, and these associations may be confounded by deprivation or aspects of deprivation (e.g. type of occupation). Nevertheless, quantifying such associations is the first step in that direction; hearing health inequalities is an emerging research area and the existing evidence on the relationship of HL with SEP and modifiable lifestyle factors is limited. We argue that there is a major public health need to assess whether HL is associated with SEP and lifestyle factors because this understanding could inform recommendations for HL preventative strategies. These could include wider implementation of interventions to promote ‘healthier lifestyles’, or governmental policies for socioeconomic equity among older people in the community.

Changes in the manuscript: We added text in the introduction (page 4, lines 34-44).

7. Methods: study population. It would be easier for readers if you cited the ELSA Cohort profile (published in The International Journal of Epidemiology).

Response: We thank the reviewer for this helpful reference, which we cited in the manuscript.

Changes in the manuscript: We cited the paper of Steptoe, A., Breeze, E., Banks, J., & Nazroo, J. (2012). Cohort profile: the English longitudinal study of ageing. *International journal of epidemiology*, 42(6), 1640-1648 on page 5, line 36 of the manuscript.

8. Outcomes: hearing loss. Lines 41-42: It is the research team who define the severity of HL, not the HearCheck device?

Response: The HearCheck™ Screener automatically generates six tones in total – one tone for each combination of frequency and level: 1 kHz at 55, 35 and 20 dB HL and 3 kHz at 75, 55 and 35 dB HL. According to the HearCheck™ Screener User Guide the lower the score of tones heard, the more likely is the patient to have a hearing difficulty and that will benefit from a hearing aid. However, these

are screening results and should not been used as audiometric assessment. So, we adhered to the categorisation that has been previously used in the literature, for the characterisation of those assessed by the same audiometric screening device (HearCheck™), which indicates the severity according to the hearing threshold in dB. Likewise, HL was defined as >35 dB HL at 3.0 kHz, in the better-hearing ear, because this is the level where intervention for HL has been shown as definitively beneficial. We cited the relevant studies in the manuscript.

Changes in the manuscript: We have now revised the Methodology section to address this specific point (page 6, lines 3-15).

9. Variables. The authors should specify when (i.e. at which wave) the variables were assessed.

Response: We revised this sentence and we now clarify that information was collected in the seventh wave of ELSA, between June 2014 and May 2015.

Changes in the manuscript: Added text on page 7, line 42 of the manuscript.

10. Statistical analysis. The authors' statement on missing data lacks detail. I find it surprising that "there were no missing values in the hearing data of the final analytical sample". Can the authors check whether some participants gave consent, did not have an ear infection, but for some reason had no usable hearing data?

Response: Our analytic cohort was composed of 8,529 individuals participating in the wave 7 of the English Longitudinal Study of Aging (ELSA), who were 50-89 years of age and whose hearing was assessed by self-report measures, that gave consent for assessment by pure-tone-audiometry and did not have any ear infection or a cochlear implant. All these 8,529 individuals (of the 9,666 initial sample in ELSA wave 7), had usable objective hearing data, measured by a qualified nurse. In total, 257 participants refused to have the assessment (the 2.6% of the full cohort of 9,666 participants). However, there was no pattern in the missing data regarding age/sex/education/occupation/income/wealth. We added text in the manuscript to increase clarity regarding the missing data.

Changes in the manuscript: Added text on page 9, lines 38-48 of the manuscript.

11. Furthermore, what were the differences (if any) between the participants at Wave 7 and those in the analytical sample? The amount of missing data may have been low (<5%) but were there any systematic differences between participants with and without missing data? Does this effect the representativeness of the findings?

Response: The amount of missing data in the sociodemographic variables was low (<5%) and without systematic differences or missing data patterns between participants with and without missing data. There were small amounts of missing data on many variables and the exposure and covariates in the main analysis were missing completely at random. Thus, records with missing data were dropped from analysis and we concluded that this did not lead to loss of precision and did not affect the representativeness of our findings.

12. Furthermore, the authors make no mention of using any weights to increase representativeness of the ELSA data. As an ageing cohort study, it is likely that the ELSA sample used in this study (without weights) is not representative but is likely to be healthier and wealthier than the older household English population. The potential bias I mentioned in the HSE study will in all likelihood be stronger with ELSA data.

Response: We thank the reviewer for his helpful comment that improved the clarity of the limitations section. The 8,249 of the 9,666 respondents in ELSA wave 7 were core members. The cross-sectional weight for ELSA wave 7 was computed only for the new entrants at wave 7, aiming to adjust for differential non-response to the self-completion questionnaire amongst respondents, and to retain the representativeness of the sample. In our analyses, we did not use the non-response statistical weights for the refreshment sample members, who were selected from HSE 2011 and 2012, which may have reduced the generalizability of our findings.

Changes in the manuscript: We added text in the limitation section, page 14, lines 35-39.

13. Finally, there is no information in the statistical analysis section to explain which measure indicates how much variance in the outcome is explained by the variables in the model (as outlined in the strengths and limitations of the study box). Presumably, this is the Pseudo R² but this needs to be clarified for the reader.

Response: We thank the reviewer for his comment. We now clarify that the variants of pseudo R squared statistics were based on the deviance of the models and used to express how much variance in the outcome is explained by the variables in each stepwise multiple logistic regression model.

Changes in the manuscript: Added text in the statistical analysis section on page 10, lines 14-18.

14. Results. The 95% Confidence Intervals should be in %s (not 0.306 to 0.337). The estimates in Table 1 should have one decimal point only (two d.p. is spurious accuracy).

Response: We revised the 95% Confidence Intervals and all the estimates in Table 1 have now one decimal point only. We would expect guidance on this from the production team, if the paper is accepted for publication.

Changes in the manuscript: Corrected on page 10 (lines 39-42) and in Table 1.

15. Table 1 should contain an explanation of how moderate and severe HL was defined. The description of the results in Table 2 is confusing: I recommend that the authors focus on the % of HL across the subgroups (as set out in the Table: hearing acuity (%)), rather than the % obese say among those with HL. Can the authors check whether they mean SD rather than SE of the mean (Table 2).

Response: We now provide definitions of HL in Tables 1 and 2, and we corrected the SE of the mean to SD. We recalculated the descriptive statistics in Table 2 and now we provide percentages per column and not per row.

Changes in the manuscript: We added a new Table 2.

16. I am confused about the differences between Table 3 and Table 4. Is it necessary to present both these analyses? Why do the authors show odds ratios in Table 3, but logistic coefficients in Table 4? Why is Table 3 sex-specific but Table 4 not so? At first glance, this is confusing. The write up of Table 4 is also very short. Adding lifestyle factors is stated to increase the variance explained: but there is no commentary on the associations between lifestyle risk factors and HL.

Response: In Table 3 we present the results of the multiple logistic regression analysis of N=8,529, aged 50-89 with HL >35 dB HL at 3.0kHz in the better –hearing ear as the dependent variable and the SEP indicators as independent variables. On the other hand, in Table 4 we wanted to further examine the association of HL with non-modifiable (age, gender), partly modifiable (education, occupation, income, wealth) and fully modifiable risk factors (high body mass index, physical inactivity, tobacco consumption and alcohol intake above the low risk level guidelines). Table 4 is therefore not sex-specific as we examined the role of gender as one of the non-modifiable risk factors and we presented the results across the four separate stepwise multiple logistic regression models.

Changes in the manuscript: We added text in the results section (page 12, lines 20-24) regarding Table 4, to increase clarity.

17. The results in Table 4 are not immediately clear (e.g. reference categories not given). For example, why are the coefficients for education negative but the coefficients for occupation positive? It would help if you coded them all the same way (e.g. comparing lower SEP to higher SEP on all four measures: as you do in Table 3).

Response: In Table 4, we coded the socioeconomic indicators of education, income and wealth in the same way we did in Table 3, having as reference the highest category (degree/higher education and highest income and wealth quintiles). Thus, the fact that the correlation coefficients for education, income and wealth are negative provides statistical evidence of the negative relationship of the variables; higher levels of education, income and wealth are associated with lower levels of HL. Moreover, in Table 4 we wanted to examine specifically the association between manual occupations with HL and its attenuation by modifiable determinants including smoking habit, which is of a higher prevalence among those that work in routine and manual occupations in England (relevant text and citation on page 12, lines 48-53). So, in the Step 2 of the relevant stepwise multiple logistic regression model, the positive coefficients for occupation indicated that a manual occupation is more likely to be associated with HL. Coding the variable in that way, allowed us to examine in Step 3 how the addition of the fully modifiable risk factors (high body mass index, physical inactivity, tobacco consumption and alcohol intake above the low risk level guidelines) modify this specific relationship.

Changes in the manuscript: We added text in the results section (page 12, lines 20-24) regarding Table 4, to increase clarity.

18. Discussion. The authors should not claim to show any evidence of the effects of SEP on HL (first sentence of the Strengths and Limitations section). As mentioned above, the authors have too little to say on the associations between RF and HL.

Response: We thank the reviewer for his helpful comment that improved the clarity of the strengths and limitations section. We have now revised the text and we present as the main strength of our study that it is the first to examine the association of four separate SEP indicators with HL among older adults in England and the role of modifiable lifestyle risk factors in these associations. To avoid

repetition please see our responses to comments 16, 17, which respond in more details to these specific points.

Changes in the manuscript: We added text in the results section (page 12, lines 20-24) and in strengths and limitations section (page 13, lines 42-47).

19. In the Discussion, the authors explain why they looked at alcohol: but then do not explain what they found and what the implications are.

Response: We now clearly state that drinking above the low risk level guidelines increased the likelihood of HL, being in line with Chief Medical Officer's Drinking Guidelines, which suggest that it is safest not to drink regularly more than 14 units per week, to keep health risks from drinking alcohol to a low level.

Changes in the manuscript: We added text in the discussion section (page 15, lines 25-31).

20. Other comments. The authors use England and UK interchangeably; ELSA is an English ageing cohort.

Response: We revised the text to make clear that ELSA study is representative of the English population aged 50 years old and above.

Changes in the manuscript: We revised the text on page 3 (line 11), page 13 (lines 5, 27, 31, 44), page 14 (line 22).

Reviewer: 2

1. In this study, the author investigated the association between socioeconomic and lifestyle factors with hearing loss in older adults, using a cross-sectional study. The notion that socioeconomic and lifestyle factors are associated with HL among older adults as strongly as core demographic risk factors is interesting. In general, the manuscript is simple and detailed, the thinking is clear. The design is reasonable, the sample size is relatively large, the indicators are described in detail, the statistical methods are simple, and the discussion is profound, which can provide suggestions for the local government to formulate health policy strategies. However, there are some minor issues that the authors should answer to improve the quality of the manuscript. The definition of hearing loss is >35dB at 3kHz in the better hearing ear. Why 35dB and 3kHz? As far as I know, 8kHz is closely related to hearing loss in the elderly. The authors mentioned that audiometry is measured at 1kHz and 3kHz. Why?

Response: We would like to thank the reviewer for the overall positive comments and the recommendations, which substantially improved our manuscript. The HearCheck™ Screener automatically generates six tones in total – one tone for each combination of frequency and level: 1 kHz at 55, 35 and 20 dB HL and 3 kHz at 75, 55 and 35 dB HL. We clarify in the Methodology section that HL was defined as >35 dB HL at 3.0 kHz, in the better-hearing ear, because this is the level where intervention for HL has been shown as definitely beneficial. For the above reason, this categorisation has previously been used in the literature for the characterisation of those assessed by the same audiometric screening device (HearCheck™). We cited the relevant studies in the manuscript.

Changes in the manuscript: We have now revised the Methodology section, to increase the clarity (page 7, lines 3-14).

2. The diabetes, hypertension, coronary heart disease and other chronic diseases among the elderly will have an impact on your results. Have you considered this part?

Response: We appreciate the reviewer's recommendation but this study was designed to examine the role of lifestyle factors and socioeconomic status. The associations observed may be partly explained by physical and/or mental health in the two groups (with and without hearing loss), and we have now added text in the conclusion part, proposing that future research in hearing health inequalities should investigate the role of the prolonged exposure to these modifiable lifestyle behaviours in the development of HL and the role of other comorbid chronic diseases among the elderly.

Changes in the manuscript: We added text on page 16, line 51.

3. I have noticed the forest graph corresponding to table 3. Why does the author choose only some of the information in each socioeconomic indicator?

Response: The reason we chose only some of the information of the Table 3 to be represented in a forest graph, was in order to show that those in a lower socioeconomic position were up to two times more likely to have HL. We wanted to emphasise that the adjusted odds of HL were higher for those with no qualifications versus those with a degree/higher education, those in routine/manual occupations versus those in managerial/professional occupations, and those in the lowest versus the highest income and wealth quintiles.

4. The author mentioned that "This study is the first that examines the effects of four different SEP indicators (education, occupation, income, wealth) to the objectively measured HL in older adults." Can you explain how these four socioeconomic indicators are selected? And other similar studies seem to have looked at these indicators. Has the author considered synthesizing them into a new socioeconomic indicator? It's probably something no one else has done.

Response: Education, occupation, income and wealth were the four selected indicators of socioeconomic position. This choice was made as these factors encompass aspects of the life-course socioeconomic stratification, according to the list of SEP indicators that Galobardes et al., (2006) proposed (relevant text and citation on page 12, lines 13-18). Moreover, we use the term socioeconomic position instead of the term socioeconomic status, to refer concisely to the components of economic and social well-being, which is linked to both childhood and adult social class position and includes both resource-based and prestige-related characteristics, which refer to the individual's rank or status in a social hierarchy (Krieger et al, 1997). We decided a priori to run separate models for the four indicators of SES rather than estimate a single model to avoid multicollinearity, as the first reviewer proposed in their study (Scholes et al., 2018). All these references were previously explored in the introduction section.

Reviewer: 3

1. This manuscript aims to examine the relationship between socioeconomic factors and hearing loss in adults aged 50-89. Additionally the manuscript aims to examine whether lifestyle factors are associated with hearing loss. To achieve these aims the authors report cross-sectional analyses using data from wave 7 of the English Longitudinal Study of Ageing. The research question addresses a gap in the literature as research on the relationship between socioeconomic and lifestyle factors and hearing loss is limited. A strength of the study is the use of a representative sample of adults aged 50 and older. I have some concerns with the analysis and conclusions drawn and have outlined these below. As a general comment, as this paper aims to examine the relationship between hearing loss and socioeconomic/lifestyle factors, the manuscript would have benefited from some discussion on the potential links between hearing loss and the socioeconomic and lifestyle factors investigated.

Response: We would like to thank the reviewer for a thorough review and detailed recommendations, which substantially improved our manuscript. We now provide more clarity on the rationale for this study in the introduction, as hearing health inequalities is an emerging research area and the existing evidence on the relationship of HL with SEP and modifiable lifestyle factors is limited. We also argue that there is a major public health need to assess whether HL is associated with SEP and lifestyle factors because this understanding could inform recommendations for HL preventative strategies such as wider implementation of interventions to promote 'healthier lifestyles' and governmental policies for socioeconomic equity among older people in the community. We also added text in the discussion section, arguing that the associations between indicators of lower socioeconomic position and hearing loss may be markers of less healthy lifestyle, which may explain the link between HL and socioeconomic and lifestyle factors investigated.

Changes in the manuscript: We added text in the introduction (page 4, lines 33-44) and the discussion (page 15, lines 25-48).

2. Gold standard pure-tone-audiometry was not available in this study but it would be helpful if the authors could include a bit of background on how the screening device used compares with standard definitions of hearing loss through audiometry.

Response: Thanks to very helpful point of the reviewer, we have now added some background on how the screening device used compares with standard definitions of hearing loss through audiometry. Previous studies have assessed the accuracy of the Siemens HearCheck™ in detecting hearing loss and compared it with pure tone air conduction averages designated as gold standard values. Fellizan-Lopez et al. (2011) found that in cases of moderate or worse hearing loss, the HearCheck™ test fulfils all criteria of high sensitivity rate, high specificity rate and high positive predictive values to be considered an accurate tool to screen for hearing loss, without the need for soundproof audiometry booths. In their study, the HearCheck™ showed high specificity rate whether the examination is done inside the soundproof booth (92.42%) or in a quiet room (95.45%). The high positive predictive value shows that among those with hearing loss based on the HearCheck™ results, the probability of having actual hearing loss is 9 out of 10, especially in cases of moderate or worse hearing loss (41dB and above air conduction average). False positive results are practically nil.

Changes in the manuscript: We added text and cited this reference in our manuscript to increase clarity regarding the reliability of HearCheck™ for determining the degree of hearing loss (page 6, lines 36-46).

3. The phrasing of the definition of moderately severe or severe HL is somewhat confusing and may benefit from dropping "(0 or 1 of the three tones at 3.0 kHz heard)".

Response: The HearCheck™ Screener automatically generates six tones in total – one tone for each combination of frequency and level: 1 kHz at 55, 35 and 20 dB HL and 3 kHz at 75, 55 and 35 dB HL. According to the HearCheck™ Screener User Guide the lower the score of tones heard, the more likely is the patient to have a hearing difficulty and that will benefit from a hearing aid. However these are screening results and should not be used as audiometric assessment and we therefore adopted the categorisation that has been previously used in the literature, for the characterisation of those assessed by the same audiometric screening device (HearCheck™), which indicates the severity according to the hearing threshold in dB. We cited the relevant studies on the page 6 of the manuscript.

4. Page 6: I would recommend the authors to separate out the definition of normal hearing from the definition of moderately severe or severe HL into a third “c” category for clarity.

Response: We thank the reviewer for his helpful comment. We now separate the definition of normal hearing from the definition of moderate severe or severe HL as the section refers to the classification of those with hearing loss. We now clarify that the ordinal variable “hearing acuity” (in better ear) was consisted of the above two categories of HL and the category of “normal hearing”, which was defined as having heard all the three tones of the hearing screening test at 3.0 kHz.

Changes in the manuscript: We revised the text on page 7 (lines 27-33).

5. The definition of occupation in the study could be clearer. The authors state “Tertiles of self-reported occupation were based on the National Statistics socio-economic classification (NS-SEC): managerial and professional; intermediate; routine and manual occupations.” which suggests occupation was defined on the level of the participant. However the authors then go on to state “This was based on the household reference person with the highest income” which suggests it was based on the household level. Please could the authors clarify whether the classification of occupation was on the individual or household level. If, on the household level, please would the authors justify why the occupation of the person with the highest income was used rather than the participant, given the proposed links between manual occupation and noise exposure raised by the authors. The author’s state: “Thus, it was possible that the household reference person was a woman” which is evident from the previous statement and therefore could be dropped.

Response: Indeed, in our analyses the occupation was defined on the level of the participant and not the household. The statement regarding the household level was mistakenly mentioned there, as it was referred to another analysis we made of the social class that was based on the household reference person, who was either the person responsible for owning/renting, or responsible for the accommodation, but actually was beyond the scope of this study.

Changes in the manuscript: We deleted text on page 7 (lines 51-55).

6. Please would the authors indicate if age was treated as a continuous or categorical variable in the regression models.

Response: We indicate now in the statistical analysis section that age was entered into the multivariable logistic regression models as a continuous variable, to maximise power.

Changes in the manuscript: Added text in the statistical analysis section on page 10, lines 11-13.

7. Tables 1 and 2 are presented with percentages calculated row-wise which is less conducive to the reader's understanding than percentages calculated column wise would be. I.e. The authors indicate that 32.14% of males and 22.25% of females have HL >35dB at 3.0 kHz. What may be more informative would be to know the proportion of individuals with HL >35dB who are male and female.

Response: We have now revised the Table 1 to show the distribution of socio-demographic characteristics of the sample (n=8,529, aged 50-89) according to hearing acuity, with percentages calculated column-wise. The proportion of men and women with HL >35 dB HL at 3.0kHz was 53.9 (1,158) and 46.2 (994), respectively. However, men were 1.5 times more likely to have moderately severe or severe HL compared to women. One in three adults aged 65-75 had hearing loss and the percentage of HL in age band 75-89 was threefold larger than in age band 50-64, as one out of every two adults aged 75-89 had HL >35 dB HL at 3.0kHz.

Changes in the manuscript: We added text on page 10 (lines 38-54) and a new Table 1.

8. It would be useful if the authors could add "better-ear" when describing hearing acuity in tables 1 and 2.

Response: Thank you, we now clarify that hearing acuity refers to the better-hearing ear.

Changes in the manuscript: We corrected the description in all Tables.

9. The footnotes of tables 1 and 2 state "Values are expressed as n (%)" but the reverse "% (n)" is actually shown in the tables. This should be corrected.

Response: We thank the reviewer for this helpful comment.

Changes in the manuscript: We corrected this statement in Tables 1 and 2.

10. The analyses and results shown in Table 4 need more explanation. Why has age (mean) been used rather than age as a continuous variable? Why is only one category of each of the lifestyle variables shown? Showing the crude and adjusted odds of each variable on hearing loss would be informative here.

Response: This was a typo and we have now corrected it. We added text to clarify that age was entered into the multivariable logistic regression models as a continuous variable. In Table 4, we coded the socioeconomic indicators of education, income and wealth in the same way we did in Table 3, having as reference the highest category (degree/higher education and highest income and wealth quintiles). Thus, the fact that the correlation coefficients for education, income and wealth are negative provides statistical evidence of the negative relationship of the variables; the increase in education, income and wealth will cause the decrease in HL. Likewise, the positive correlation coefficients for occupation indicated that the increase in occupation (to manual occupations) cause the increase in HL. Thus, the purpose of Table 4 was to present the results across the four separate stepwise multiple regression models, that show the association of HL with non-modifiable (age, gender), partly modifiable (education, occupation, income, wealth) and fully modifiable risk factors (high body mass index, physical inactivity, tobacco consumption and alcohol intake above the low risk level guidelines), respectively.

Changes in the manuscript: We added text on page 10, lines 11-13 and we did the correction for age in Table 4.

11. The discussion introduces figure 2 which would be better introduced in the results section. Additionally the discussion states that the difference in prevalence “begins” at 50-64 whereas that was the earliest age band included in these analyses. Therefore a more accurate description would be to state that differences in hearing loss prevalence between males and females were observed across all age bands investigated. The authors state that “male sex is not a consistent risk factor in studies” and cite Cruickshanks et al 1998 as a reference for this statement. However, in the Cruickshanks et al study hearing loss was greater for men than women and remained so after adjustment for demographic factors, noise exposure, and occupation. Male sex is often cited as a consistent risk factor for hearing loss (e.g. Hoffman et al, 2016; Cruickshanks et al, 2015; Lin et al, 2011)

Response: We thank the reviewer for this comment and the useful references, which we found particularly helpful for improving our discussion section. We revised the text to improve clarity and we now argue that the differences in modifiable lifestyle factors that were revealed in the stepwise regression models may finally explain why the male sex is often cited as consistent risk factors for hearing loss, as you correctly mentioned. We therefore propose now that this finding leads to the exploration of modifiable determinants that are common in both genders, paving the way for interventions to improve the population’s hearing health.

Changes in the manuscript: We revised the text in results section (page 12, line 32-40) and in discussion section (page 16, lines 5-11) and we added the above references in our manuscript (references No 35, 36, 37).

12. The authors state in the abstract and discussion that “socioeconomic and lifestyle factors” (abstract) and “several modifiable lifestyle factors” (discussion) are associated with hearing loss “as strongly” as the demographic risk factors of age and sex. However, I do not think such a conclusion is justified based on the results presented. Table 4 illustrates a small increase in the variance explained by socioeconomic factors compared to age and sex only. When several lifestyle factors are added in the model the variance explained does increase but the collective effect of all the lifestyle factors examined is still less than that of the demographic risk factors. The authors should discuss how their findings compare with Cruickshanks et al, 2015 who also investigated the impact of occupation and some lifestyle factors (BMI, alcohol, smoking) on hearing loss.

Response: We revised the text in results and discussion section according to the reviewer’s suggestion. We argue now that socioeconomic and several modifiable lifestyle factors (such as high body mass index, physical inactivity, tobacco consumption and alcohol intake above the low risk level guidelines) are associated with the likelihood of HL as strongly as well-established demographic factors such as age and gender HL. According to the stepwise multiple regression models, the addition of lifestyle factors attenuated significantly the association between the HL and SEP indicators and in total the addition of SEP and lifestyle factors in the regression models explained another 10 to 15% of the variance in the likelihood of HL. The total variance explained in the overall models containing demographic factors, SEP and lifestyle factors ranged between 25 and 27%. This finding suggests that SEP and lifestyle factors have an equal contribution to HL as age and gender. We also added text in the discussion to compare with Cruickshanks et al, 2015 who addressed hearing loss in a younger population-based sample (aged 18 to 74 years) of Hispanics/Latinos, and included in the multivariable-adjusted model body mass index, smoking, and alcohol but they found that these factors were not significantly associated with hearing impairment. That may reveal that hearing loss in older

population (e.g. 50 years and above) is probably associated with different risk factors or even with the cumulative effect of the socioeconomic and lifestyle risk factors across the life-course.

Changes in the manuscript: We added text on page 12, (lines 32-40), page 15 (lines 25-48) and a new reference (No 36).

Reviewer: 4

1. The present manuscript describes an exciting study on possible associations between socioeconomic and lifestyle factors and HL among older adults in England. The study population, n= 8,529 participants, was available from an ongoing prospective English longitudinal study of aging focussing on social, wellbeing and economic circumstances. The study shows that socioeconomic and modifiable lifestyle factors are associated with HL among older adults as strongly as core demographic risk factors, such as age and gender. However, the analysed factors explained less than 30% of the variance for the occurrence of HL, which means that there are additional major factors associated with HL in older adults, which still has to be investigated and identified. One such important factor, which was not registered in the study population was possible influence of exposure to noise. Despite these shortcomings this study supports the suggestions that HL may be prevented or improved by public health policies and interventions. The study is well designed and well performed and the results original. I am ready to recommend this manuscript for publication in BMJ Open., but before that I would like to have the authors to answer the comments and questions below.

Response: We would like to thank the reviewer for the overall positive remark and the comments, which substantially improved our manuscript. Indeed the occupational/social noise exposure is an important factor, with a damaging effect on hearing. However, we were not able to examine this association with HL, as the ELSA dataset does not record any relevant information, which could presumably contribute to a greater percentage of the explained variance for the occurrence of HL. However, we examined the role of manual occupations in the relationship of socioeconomic position with HL, as the manual jobs tend to be those with a higher level of (occupational) noise exposure and we mention the above limitation in the relevant section (page 14, lines 22-35).

2. "Older adults" in this study are aged 50 to 89 years of age. Which is the background to this definition of older adults and not e.g. 65 to 90? In Table I the HL is shown for 3 age groups; 50-64, 65-74 and 75-89. I should have been of interest to have these different age groups for most variables, not the least concerning the modifiable lifestyle factors. Similarly, I have difficulties to see the relevance of Retirement status in Table 1 if not correlated to age groups within the age span of 50-89.

Response: We have now revised the text to make clear that the ELSA study is representative of the English population aged 50 years old and above. We now mention in the Methodology section that age was categorised into three groups (50-64, 65-74, 75-89) to allow for a comparison with Benova et al. (2015), who examined the association of socioeconomic position with hearing difficulty in ELSA wave 2.

Changes in the manuscript: We added text in the Methodology section to improve clarity (page 8, lines 23-28) and we added a new Table 1, where percentages are calculated column-wise, so as to show the difference of HL percentages among those who are retired or not.

3. The objective measurement of hearing acuity was performed by a handheld screening device, HearCheck™. In p 6 it is described that the hearing test is performed with a series of three sounds at 1.0 kHz and 3.0 kHz. For the determination of the hearing loss only measurements at 3.0kHz are used and described in text. Why including 1.0 kHz in the test situation and then not using it to characterize the HL?

Response: We clarify in the Methodology section that HL was defined as >35 dB HL at 3.0 kHz, in the better-hearing ear, because this is the level where intervention for HL has been shown as definite beneficial. For the above reason, this categorisation has previously been used in the literature, for the characterisation of those assessed by the same audiometric screening device (HearCheck™). We cited the relevant studies.

Changes in the manuscript: We have now revised this section to increase the clarity (page 7, lines 3-14).

4. The HearCheck™ has been used in some other scientific studies. How reliable is the present method for determining the degree of hearing loss?

Response: The HearCheck™ Screener automatically generates six tones in total – one tone for each combination of frequency and level: 1 kHz at 55, 35 and 20 dB HL and 3 kHz at 75, 55 and 35 dB HL. Previous studies have assessed the accuracy of Siemens HearCheck™ in detecting hearing loss and compared it with pure tone air conduction averages designated as gold standard values. Fellizan-Lopez et al. (2011) found that in cases of moderate or worse hearing loss, the HearCheck™ test fulfils all criteria of high sensitivity rate, high specificity rate and high positive predictive values to be considered an accurate tool to screen for hearing loss, without the need for soundproof audiometry booths. In their study, the HearCheck™ showed high specificity rate whether the examination is done inside the soundproof booth (92.42%) or in a quiet room (95.45%). The high positive predictive value shows that among those with hearing loss based on the HearCheck™ results, the probability of having actual hearing loss is 9 out of 10, especially in cases of moderate or worse hearing loss (41dB and above air conduction average). False positive results are practically nil. The accuracy rate of the Siemens HearCheck™ inside the soundproof audiometry booth and in a quiet room were 82.5% and 84% respectively, for all levels of hearing loss.

Changes in the manuscript: We added text and cited this reference in our manuscript to increase clarity regarding the reliability of HearCheck™ for determining the degree of hearing loss (page 6, lines 36-46).

5. p 7 – When it comes to categories of some variables there are difficulties to convert e.g educational levels to those of other countries. How should e.g. A level and O levels CSE be translated to a more international language? And what stands CSE for?

Response: We thank the reviewer for this comment. We could not change the name of the categories as they refer to the National Qualifications Framework (NQF). However, we now added text to increase clarity, indicating that CSE means “Certificate of Secondary Education” and A level refers to Level 3 Qualification.

Changes in the manuscript: Added text on page 7, lines 46-47.

6. p 9 – last sentence – HL in various age groups is described and has been commented upon above. However, though only used in relation to HL the subdividing in groups should have been mentioned in the Methodology section.

Response: We now mention in the Methodology section that age was categorised into three groups (50-64, 65-74, 75-89), to allow for a comparison with Benova et al. (2015), who examined the association of socioeconomic position with hearing difficulty in ELSA wave 2.

Changes in the manuscript: We added text in the Methodology section (page 8, lines 23-28).

VERSION 2 – REVIEW

REVIEWER	Shaun Scholes University College London, UK
REVIEW RETURNED	23-Jul-2019

GENERAL COMMENTS	I have read both the response to reviewer comments and the revised manuscript. The paper is much improved. A few comments are: Strengths and Limitations: prevalence is a better term than occurrence. Methods: the authors should clarify that the Benova study analysed self-reported hearing difficulty. Table 1: I disagree with another reviewer of this manuscript who suggested presenting column rather than row %s. Firstly, the row percentages would be useful for other researchers doing a meta-analysis. Secondly, at the very least, the authors should clarify in a note to the Table that these are column rather than row %s. (Plus the figure for men should be 59.3 not the 53.9 as quoted in the text). Actually, Figure 2 shows HL by age and gender. But from the manuscript it is not clear whether these are prevalence estimates or are estimates based on the models set out in Table 4. If the former - which is my guess - then this information may best be shown before the model results. Discussion The authors mention that they did not use weights for refreshment sample members. It would be better if they stated that they did not use any weights at all: meaning that the remaining ELSA sample members at Wave 7 could be healthier on average than the population: potentially resulting in an underestimation of relationships. These issues for example are discussed in the limitations section of the paper by Lasselle et al: Camille Lassale, G David Batty, Andrew Steptoe, Dorina Cadar, Tasnime N Akbaraly, Mika Kivimäki, Paola Zaninotto, Association of 10-Year C-Reactive Protein Trajectories With Markers of
---

	Healthy Aging: Findings From the English Longitudinal Study of Aging, The Journals of Gerontology: Series A, Volume 74, Issue 2, February 2019, Pages 195–203 Overall There are still numerous typos which are very frustrating to read. For example, in the methods ELSA is referred to as a study of aging rather than ageing. I encourage the authors to replace the word "effect" with association as is recommended in other journals. I will recommend acceptance but i hope my comments are taken on board.
--	--

REVIEWER	Sten Hellström Dept of Audiology and Neurotology Karolinska University Hospital/Solna SE 171 76, Stockholm, SWEDEN
REVIEW RETURNED	06-Aug-2019

GENERAL COMMENTS	The authors have read the reviewer’s questions and comments and carefully revised the manuscript according to the suggestions. The manuscript is acceptable for publication in its present form.
---

VERSION 2 – AUTHOR RESPONSE

Reviewer: 1

1. Strengths and limitations: prevalence is a better term than occurrence.

Response: We have revised the text accordingly.

2. Methods: the authors should clarify that the Benova study analysed self-reported hearing difficulty.

Changes in the manuscript: We revised the sentence on page 8 (line 20) to clarify that.

3. Table 1: I disagree with another reviewer of this manuscript who suggested presenting column rather than row %s. Firstly, the row percentages would be useful for other researchers doing a meta-analysis. Secondly, at the very least, the authors should clarify in a note to the Table that these are column rather than row %s. (Plus the figure for men should be 59.3 not the 53.9 as quoted in the text). Actually, Figure 2 shows HL by age and gender. But from the manuscript it is not clear whether these are prevalence estimates or are estimates based on the models set out in Table 4. If the former - which is my guess - then this information may best be shown before the model results.

Response and changes in the manuscript: We thank the reviewer for these points. We recalculated the percentages in Table 1 so as to indicate correctly the sum of moderate and moderately severe or severe HL for men and women, respectively and we revised the sentences on page 10 (line 35). We clarified in a note to the Tables 1 and 2 that the “values are expressed as column % (N) unless

otherwise is indicated". Moreover, we added a note in Figure 2 to clarify that the percentages refer to the prevalence estimates of total numbers for men and women (males: N= 3,728, females: N=4,801).

However, Figure 2 shows that the rate of deterioration of hearing acuity as age increases was similar between each age band and nearly to 60% in both genders. This highlights the importance of exploring modifiable determinants in both genders. The numbers refer to prevalence estimates for men and women and we added a clarification note in Figure 2.

4. Discussion. The authors mention that they did not use weights for refreshment sample members. It would be better if they stated that they did not use any weights at all: meaning that the remaining ELSA sample members at Wave 7 could be healthier on average than the population: potentially resulting in an underestimation of relationships. These issues for example are discussed in the limitations section of the paper by Lassalle et al: Camille Lassalle, G David Batty, Andrew Steptoe, Dorina Cadar, Tasnime N Akbaraly, Mika Kivimäki, Paola Zaninotto, Association of 10-Year C-Reactive Protein Trajectories With Markers of Healthy Aging: Findings From the English Longitudinal Study of Aging, *The Journals of Gerontology: Series A*, Volume 74, Issue 2, February 2019, Pages 195–203

Changes in the manuscript: We revised the sentence on page 14 (line 11) to clarify that.

5. There are still numerous typos which are very frustrating to read. For example, in the methods ELSA is referred to as a study of aging rather than ageing. I encourage the authors to replace the word "effect" with association as is recommended in other journals.

Changes in the manuscript: We deleted the word 'effect', corrected the highlighted typos and proof-read the manuscript throughout.